# A Review of Legal Regulation of Religious Slaughter in Australia: Failure to Regulate or a Regulatory Fail?

**DOI:** 10.3390/ani10091530

**Published:** 2020-08-30

**Authors:** Jessica Loyer, Alexandra L. Whittaker, Emily A. Buddle, Rachel A. Ankeny

**Affiliations:** 1School of Humanities, The University of Adelaide, Adelaide, SA 5005, Australia; jessica.loyer@adelaide.edu.au (J.L.); emily.buddle@adelaide.edu.au (E.A.B.); rachel.ankeny@adelaide.edu.au (R.A.A.); 2School of Animal and Veterinary Sciences, The University of Adelaide, Roseworthy Campus, Roseworthy, SA 5371, Australia

**Keywords:** religious slaughter, halal, kosher slaughter, legislation, Australia

## Abstract

**Simple Summary:**

Religious slaughter has recently attracted public attention as a result of media portrayal of several high-profile Australian and international events. The requirements of domestic religious slaughter practice, including animal welfare provisions, appear to be poorly understood by the Australian public. This paper summarizes the welfare science and regulatory framework around halal and shechita slaughter in Australia. Current knowledge on public viewpoints on these practices is examined, and areas for future social science research are proposed. In spite of wide-ranging and extensive animal welfare protection being provided by the law, we propose that the complexity of the legislative arrangement reduces transparency and undermines the strength of protection to animals provided by law. Avenues for legal reform are proposed. There is also a need for more active public engagement to increase community knowledge about religious slaughter practices, and to counter Islamophobia and anti-Semitic attitudes.

**Abstract:**

While religious slaughter is not a new practice in Australia, it has recently attracted public concern regarding questions of animal welfare following unfavourable media coverage. However, the details of religious slaughter practices, including related animal welfare provisions, appear to be poorly understood by the Australian public, and no existing literature concisely synthesises current regulations, practices, and issues. This paper addresses this gap by examining the processes associated with various types of religious slaughter and associated animal welfare issues, by reviewing the relevant legislation and examining public views, while highlighting areas for further research, particularly in Australia. The paper finds shortcomings in relation to transparency and understanding of current practices and regulation and suggests a need for more clear and consistent legislative provisions, as well as increased independence from industry in the setting of the standards, enforcement and administration of religious slaughter. A starting point for legal reform would be the relocation of important provisions pertaining to religious slaughter from delegated codes to the responsible act or regulation, ensuring proper parliamentary oversight. In addition, more active public engagement must occur, particularly with regard to what constitutes legal practices and animal welfare standards in the Australian context to overcome ongoing conflict between those who oppose religious slaughter and the Muslim and Jewish communities.

## 1. Introduction

Meat is central to Australian cultural identity and economic prosperity. Australia has earned its reputation as a heavy meat-eating nation, with average annual consumption estimated at over 110kg per person [1], around three times higher than the global average. Meat production for export is an important industry, with Australia leading the world as the largest sheep and goat meat exporter and third largest beef exporter [2]. Australia is also an increasingly religiously diverse nation. The predominance of Christianity has significantly decreased from 88% of the population in 1966 to 52% in 2016, while Islam (2.6%), Buddhism (2.4%), Hinduism (1.9%), Sikhism (0.5%), and Judaism (0.4%) now constitute significant minorities; 30.1% of the population report having no religion [3]. Both Islam and Judaism have requirements for slaughter according to religious law, creating domestic and significant export demand for meat produced in certain ways [4].

The demand for kosher meat—meat slaughtered in accordance with Jewish dietary law (kashrut)—is relatively small in Australia. About 0.4% of the population (91,022 people), identified their religion as Jewish in 2016, although this is considered to be an underestimation [5]. Following kashrut is a relatively low priority for most Australian Jews, ranking second to last on a list of 18 practices in a large national survey, highlighted by just 29% of Australian Jews reportedly buying only kosher meat for the home. [6]. 

Demand for halal meat, on the other hand, is growing alongside the increasing Muslim population. Islam is now the second largest religion in Australia, with 604,240 Australians identifying Islam as their religion in 2016. This represents an increase in population share from 2.2% in 2011 to 2.6% in 2016 [3]. Similar to the Jewish community, Muslims are largely concentrated in Melbourne and Sydney, but numbers are growing in all capital cities. Although scholarly literature on the religiosity of Australian Muslims is scant, we know there is some diversity in interpretation regarding what makes meat halal, as more generally this is subject to interpretation by local religious authorities. While some Muslims require meat to be slaughtered by a Muslim and/or certified as halal, others believe that meat slaughtered by other “people of the book” (i.e., Christians and Jews) is acceptable and will therefore purchase meat from ordinary butchers or supermarkets [7]. 

As the Muslim population grows worldwide, the concept of Islamophobia has become an increasingly important term in academic, political, and media discourse. While a 2018 survey found that only 10% of the Australian population are highly Islamophobic and 20% are undecided, most Australians felt more “social distance”, or less affective closeness and intimacy, in relation to Muslims than to members of other religions. In short, while most Australians are not explicitly anti-Muslim, they do harbour more prejudice and feel less sympathy towards this group [8]. This negative sentiment may be exacerbated by media narratives focusing on violent extremism, Muslim immigrants and asylum seekers, and reports of Islamophobic incidents abroad [8], as about half of non-Muslim Australians rely on mainstream news media as a primary source of information about Islam and Muslim people [9].

While religious slaughter is not a new practice in Australia, it has recently attracted public concern regarding questions of animal welfare following unfavourable media coverage [10]. However, the details of religious slaughter practices, including animal welfare provisions, are poorly understood by the Australian public [11], and no existing literature concisely synthesises current regulations, practices, and issues. This paper addresses this gap by examining the processes associated with various types of religious slaughter and animal welfare issues that arise, as well as briefly addressing public views and highlighting areas for further research, particularly in Australia.

## 2. The Welfare Science

### 2.1. Traditional Slaughter Methods

The prescribed traditional method of Islamic slaughter requires a single knife cut to the neck to sever the jugular vein, carotid artery, trachea, and oesophagus [12]. Using this method without pre-stunning raises welfare issues, including the stress of restraint, the painful nature of the cut, potential distress caused by bleeding, and the time to onset of insensibility [13,14,15]. The Judaic shechita method for slaughtering of certain mammals and birds for food, according to the set of dietary laws, called kashrut, is similar. A trained slaughterman makes a transverse cut across the neck with a knife [16], the blade of which is no less than double the width of the neck of the animal to be slaughtered. The knife is inspected for sharpness after each cut [17]. After slaughter, the carcass is examined and forbidden tissues, such as blood, are rejected, along with areas containing defects, such as haemorrhages [16,18].

In both halal and shechita slaughter, appropriate animal restraint is critical in maximising animal welfare, improving the quality of the meat product, maintaining adherence to slaughter rules, and safeguarding operator safety [19,20]. If animals panic due to inadequate restraint, the accuracy of the incision may be impaired [21]. For halal slaughtering, animals should be positioned on their left flank, facing in the direction of the Kaaba sacred building at Mecca at the onset of incision [16]. According to the verses of the Hadiths (a major source of religious law and moral guidance in Islam, second only to the authority of the Qurʾān), animals should only be shackled and hoisted after bleeding once the animal has lost consciousness [22]. However, it appears that some countries may use stressful restraint methods, such as shackling or hoisting of conscious animals by a rear leg [15]. Aside from the suffering that this may cause, injuries arising from these methods may mask the animal’s reaction to the throat cut and are likely objectionable to religious proponents [15].

The soft tissues of the neck are extensively innervated by nerve fibres, which will fire when transected, leading to pain and potential psychological shock when animals are not stunned prior to incision [23]. Various studies utilising the measurement of behaviours, EEG, cranial–nerve activity, and brain neurotransmitter release [3,24,25,26] have shown that unconsciousness, with the accompanying loss of ability to feel pain and experience distress, is not achieved immediately following incision [23] and thus time to unconsciousness or insensibility is an important parameter in evaluating welfare impact. Studies suggest that this period ranges between about 3 and 60 or more seconds, dependent on species (see [23]). In animals that take longer than 60 s to lose consciousness, it is likely that a swelling of the cut ends of the carotid arteries may have occurred, creating a false aneurysm; it has been proposed that the likelihood of this occurring can be reduced by applying the neck cut in a more cranial location [27,28].

A further welfare concern associated with slaughter without stunning is the aspiration of blood into the lungs [29]. Gregory et al., in 2009, performed post-mortem respiratory tract surveys of cattle following non-stun halal slaughter and discovered that between 58–69% of animals had blood in the trachea and upper airways [29]. This led the researchers to conclude that airway irritation by blood could contribute to distress prior to insensibility onset. The use of a more cranial cut, as described earlier, could mitigate distress from inhaled blood by deafferenting the respiratory tract, reducing sensory signal transmission evoked by tract blood [27]. Conversely (see [27]), some scientists argue that suffering is not caused by blood in the lungs or airways, since signals are carried by vagal afferents [30], which are severed during the neck cut. It is worth noting that the study by Gregory et al., in 2009 [29], also demonstrated that animals stunned by captive bolt, causing immediate cessation of breathing, had blood in their tracheas. The incidence of this was similar to those killed using the shechita method. Furthermore, welfare concerns surrounding the aspiration of blood have tended to focus on animals held in an inverted position during the cut [31], yet this study indicates that the same concerns exist for cattle slaughtered in an upright orientation [29]. 

### 2.2. Stunning

Stunning is defined as a process used to cause loss of consciousness and sensibility without pain, performed before or at the same time as animals are killed [32]. The method used should produce rapid insensibility and unconsciousness until death supervenes. Stunning methods were initially developed as animal immobilisation techniques to ensure personnel safety [33]. However, in recent years, considerable focus has been placed on the benefits of pre-slaughter stunning from an animal welfare perspective. As is the case with conducting slaughter without stunning, a key factor in assessing the welfare impacts of stunning methods and associated slaughter techniques is the time to onset and duration of unconsciousness [34], alongside the practicality of being able to cause death within this period. Stunning methods most commonly used include mechanical stunning (captive bolt), application of an electrical current through the head of the animal, or gaseous atmospheric stunning, which involves the immersion of the animal in a mixture of gasses which, in combination, creates an anoxic environment [34]. It is worth noting that stunning techniques themselves may have a negative welfare impact—for example, the occurrence of accidental pre-stun electric shocks or aversion to gaseous atmospheres (see [35] for review). Therefore, in considering the overall impact of religious slaughter methods on welfare, a comparison should be made between non-stun, halal-compliant pre-stun methods, and the remaining secular methods [35].

Since the 1980s, a range of Islamic authorities globally have accepted pre-slaughter stunning, provided it meets pre-requisite conditions based on the interpretation of Islamic law. These pre-requisites include: (1) stunning equipment must be under the control of a Muslim supervisor and should be monitored by a halal certification authority or Islamic authority; (2) the stunning must be reversible or temporary so as not to kill or cause permanent injury to the animals; (3) stunning equipment has not been in contact with pigs or other haram (forbidden) ingredients, such as alcohol [36]. Based on the need for the stun to be reversible, only a few recognised stun methods are likely to be halal compliant. 

Non-penetrative stunners used in mechanical stunning produce a blow to the animal’s skull and result in a reversible stun, and are acceptable to Islamic authorities, provided the skull is confirmed to be intact following skinning [36]. Electrical stunning leads to an unconscious state by generating an epileptiform seizure, which renders the animal insensible to pain [37]. Different electrode placement results in varied stun outcomes: placement on the body (rather than the head) will induce cardiac fibrillation and immediate cardiac arrest, which means that this stunning method does not meet the halal criteria of a reversible stun [38]. However, the alternate placement of electrodes for a head-only stun, resulting in seizure activity, may be acceptable. Gaseous stunning methods are mainly used for pigs and poultry, [35] and are controversial. In spite of the potentially reversible nature of these methods, they are not permitted for the halal slaughter of poultry (pork, of course, being declared haram as part of Islamic law) [12]. The World Halal Council cites gaseous methods as cruel, with the aim being only to kill the animal. Consequently, since Islam forbids the eating of carrion, the method is regarded as non-Islamic [39]. 

Stunning is not accepted for Shechita slaughter [16]. The reasons for rejection are a belief that the non-stun, traditional method is superior, painless, and causes instantaneous insensibility, whereas stunning causes damage to carcasses and may inadvertently kill the animal [16]. 

## 3. Legal Regulation of Religious Slaughter in Australia

Under the Australian Constitution [40], the federal government cannot favour or fund any religion, nor prohibit the free exercise of religion, and hence is a secular state [41]. Thus, the direct banning of halal or kosher slaughter practice would likely breach s116 of the Constitution. Religious slaughter is therefore accommodated by the provision of legal protections for non-standard slaughter methods which occur as part of religious practice [42]. There are also significant economic benefits gained from the halal-certified food market, with spending in the Australian domestic market valued at AUD 1.7 billion annually (2019 data). Globally, Australia is the fourth largest exporter of halal food and beverages to Organisation of Islamic Cooperation Countries with an annual export value of AUD 7.8 billion [43].

Whilst there is a dearth of empirical data, there is a suggestion, based on limited research [11] and online sources, that the Australian public may misunderstand the process of religious slaughter and the law surrounding it in the Australian states and territories [44,45]. Many believe that religious slaughter always or mostly occurs without pre-stunning, and hence that these practices raise considerable concerns with regard to animal welfare [44]. It is not difficult to see how public confusion surrounding the relevant regulation has arisen: a cursory glance at the legal framework in this area reveals reliance on a variety of interconnected pieces of delegated legislation, with a range of compliance agencies both cross and intra-jurisdictionally. The following summarises the relevant regulation as it pertains to meat production for human consumption in order to highlight how a relative lack of transparency may be contributing to these misunderstandings. 

### 3.1. Domestic Abattoirs 

Australia is a Federation of eight states and territories. Since neither animal welfare nor domestic meat production fall under a “head of power” for Commonwealth legislation under s51 of the Commonwealth of Australia Constitution Act (“Constitution”) [40], they are residual powers and remain the domain of the states. Consequently, there are eight territorial jurisdictions to be considered in describing the regulation of religious slaughter. These Acts empower responsible officials to make subordinate (delegated) legislation covering issues that fall within the scope of the Act. Delegated legislation includes regulations, ordinances and codes of practice/standards. These documents contain more detail than the primary act but are usually associated with reduced penalties in the event of breach of a provision and may not be subject to parliamentary oversight. In general, the framework for regulation across the states and territories is similar, with minor differences in the method of inclusion of codes and in penalty provisions. Thus, the following presents a broad summary, rather than detail on a state-by-state basis. 

The state’s legislative arrangement provides for overarching acts, the primary objects of which are to ensure food/meat is safe for human consumption by providing for the establishment of food safety schemes [46,47,48,49,50,51,52]. For example, the Victorian Meat Industry Act 1993 has as its purposes:(a)To set standards for meat production for human consumption and pet food;(b)To set up a licensing and inspection system and a mechanism for adopting and implementing quality assurance programs to ensure that those standards are maintained;(c)To enable the regulation of meat transport vehicles;(d)To establish the Victorian Meat Authority to operate that licensing and inspection system and arrange for the adoption and monitoring of quality assurance program.

The states vary as to whether the overarching act is focused on meat alone or relates to food products more generally. Where there are separate acts pertaining to the meat industry, there are commonly “Food Acts” and associated delegated legislation which also need to be complied with by abattoirs, since their primary goal is to ensure food safety and prevent misleading conduct in relation to food. The primary focus here is on the act that empowers the governing Standard in relation to humane slaughter. Figure 1 illustrates the general ‘hierarchy’ of primary and secondary legislation as it pertains to religious slaughter. 

Key features of the acts are that they enable delegated legislation, such as regulations and standards, provide the offences and associated penalties, and establish enforcement agency appointment and their powers. Detailed provisions in relation to food safety schemes and licensing are provided for by the relevant regulations [53,54,55,56,57,58,59,60]. The incorporation of the Commonwealth Australian Standard for the Hygienic Production and Transportation of Meat and Meat Products for Human Consumption (AS4996:2007)—(“The Meat Standard”) [61] into the legal regime is through the Regulations [54,56,57,58,59,60,62] or referral in licensing conditions [55]. This document, and similar versions for poultry [63] and other types of meat [64,65], provide the primary guidance on the regulation of religious slaughter in Australia. 

The Standard [61] provides that:
7.9 Animals are slaughtered in a way that prevents unnecessary injury, pain and suffering to them and causes the least practicable disturbance.7.10 Before sticking commences, animals are stunned in a way that ensures the animals are unconscious and insensible to pain before sticking occurs and do not regain consciousness or sensibility before dying.7.11 Before stunning commences, animals are restrained in a way that ensures stunning is effective.


It is clear from the above that the status quo is that slaughter animals are rendered unconscious prior to skin penetration and major blood vessel trans-section (stick). Hence the public should be reassured that the majority of Australian halal meat has come from animals treated no differently at slaughter than those destined for the non-halal market, with the exception of the religious requirements around utterances and personnel. As stunning is defined as “a procedure for rendering an animal unconscious and insensible to pain”, the Standard is silent on specific methods of stunning, hence allowing for discretion as to whether reversible methods are used to satisfy recent interpretations of Islamic law. 

The exemption for religious slaughter is:
7.12 (1) This provision only applies to animals killed under an approved arrangement that provides for their ritual slaughter involving sticking without prior stunning.(2) An animal that is stuck without first being stunned and is not rendered unconscious as part of its ritual slaughter is stunned without delay after it is stuck to ensure it is rendered unconscious.

Clause 7.12 is controversial. Animal protection groups [66,67], the Greens, and some Meat Industry Groups [68], have called for pre-slaughter stunning to be mandatory rather than the stunning process occurring after the animal is stuck. A national guideline, Ritual Slaughter for Ovine (Sheep) and Bovine (Cattle), provides further detail around 7.12. (2) For cattle, the stunning is to take place immediately after the throat cut, with two slaughtermen present so that one performs the cut and the other stuns. Stunning is usually mechanical, done by captive bolt and the animal is to be restrained in an upright position. For sheep, stunning is not required after sticking unless the animal is distressed or does not rapidly lose consciousness [67]. The reason for this inter-species difference in requirement for post-stun is due to anatomic differences in the vasculature to the brain [69]. Vertebral arteries in cattle provide an alternate route for blood to reach the brain, and since not severed by the neck cut, prevent immediate insensibility through the cut alone. It has traditionally been believed that similar concerns did not exist for sheep and goats and hence there was no need for post-cut stun. However there is mounting anecdotal evidence that time to insensibility is protracted in a number of sheep and goats possibly because of the same alternative arterial pathway, and therefore further scientific research and review of this guidance is required [70]. It is worth noting that some Muslims have concerns about the effects of a post-cut stun on effective bleeding out [12,71,72], although these effects are disputed in the limited available scientific data on this issue [73].

It is not clear how many animals within Australia are slaughtered under the exemption for religious slaughter and, therefore, without stunning, although Animals Australia estimated that this may include 250,000 sheep annually [68]. For the domestic halal market, this exemption is only needed for a “conservative minority” [68]. Conversely, kosher meat production relies entirely on the operation of this clause, since pre-stunning is not accepted. The market for kosher food in Australia is estimated to be small, perhaps servicing an estimated 30,000 people, and the vast majority of kosher animal slaughter is said to occur in NSW and Victoria [74]. Based on information obtained by Animals Australia under a Freedom of Information request, there are eight abattoirs nationwide with an approved exemption for religious slaughter, with two each in NSW and SA, and four in Victoria [75]. This lack of transparency regarding the extent of non-stun slaughter practice has been noted not just by animal protection groups, but by those involved in the meat industry [68], and arguably does little to dissuade broader concerns in Australia surrounding religious slaughter. 

It should be noted that the law plays an additional role in safeguarding animal welfare at the time of slaughter through the operation of the state and territory Animal Protection Acts [76,77,78,79,80,81,82,83], hereafter termed "Animal Welfare Acts". Provisions under these acts have been used in prosecuting various high profile abattoir cruelty cases [84,85]. These acts provide an avenue for prosecuting offences of animal cruelty (as defined in the act) should it occur in commercial slaughtering establishments. They can also provide protection from prosecution for those carrying out religious slaughter as part of an approved exemption. Prima facie, all states and territories have provisions to ensure the legality of religious slaughter through a range of different mechanisms (Table 1). For example, consider the Prevention of Cruelty to Animals Act 1979 (NSW) as a representative example. Section 5(3) (b) states:
A person in charge of an animal shall not fail at any time where pain is being inflicted upon the animal, to take such reasonable steps as are necessary to alleviate the pain.

However, there is a later defence at section 24 (c) so the person is not guilty if the act or omission was in the course of, and for the purpose of, destroying the animal, or preparing the animal for destruction: (i) in accordance with the precepts of the Jewish religion or of any other religion prescribed for the purposes of this subparagraph, or (ii) in compliance with any duty imposed upon that person by or under this or any other act.

It is also noteworthy that standard slaughter practice is also listed as a defence to cruelty at S24(b) (i), where no offence is committed whilst destroying the animal for the purpose of producing food for human consumption, provided that no unnecessary pain was inflicted upon the animal. This clause highlights legal recognition of the inevitable animal welfare impact of any method of humane killing or slaughter. Many of the Animal Welfare Acts use terminology, such as unnecessary (e.g., [76]), unjustifiable [81], or unreasonable [76,81], to define offences and defences. These terms are often poorly defined and it remains the role of the courts to interpret and apply the terms based on expert witness testimony and precedents [86,87]. Such terminology remains controversial in animal law jurisprudence with the suggestion that it allows legislators to justify suffering that is unavoidable in practice, and hence allows the current status quo with respect to farming practices, and other use of animals in society, to continue [86].

Alongside the Animal Welfare Acts, there is a Model Code of Practice for the Welfare of Animals: Livestock and Poultry at Slaughtering Establishments [88], which contains some information relevant to religious slaughter. There are a few points to highlight regarding the Model Code of Practice. The language of this code is generally not worded as a directive and may be construed as ambiguous. For example, the use of the words “must” and “should” abound. Common law jurisdictions tend to avoid the use of "should" or "shall" in legal drafting since these terms can introduce ambiguity. These terms may not confer an obligation, and can be variably construed to signal the future tense (may or will), especially to Australian, British, and Canadian legal drafters [89]. For example, it is stated at 2.6.1.6 that “stunning for religious slaughter *should* be encouraged”. It is not clear whether these documents are actually in opposition, or if this is just an ambiguity of language. This appears in direct contravention of the Australian Standard adopted under the Meat/Food Acts which requires stunning unless an exemption has been obtained. This code has varying legal force dependent on jurisdiction and is generally referred to in relation to the Animal Welfare Acts. For example, in South Australia this code is prescribed under Schedule 2 of the Regulations and, therefore, breach of a provision makes the offender liable to a penalty [62]. In most jurisdictions this code is not compulsory and, therefore, there is no direct penalty in the event of a breach, but non-compliance is admissible evidence in court in the event of an animal cruelty matter [90]. This inconsistency does not appear to have been tested in the courts; however, it would be assumed that, in states where the code is prescribed, the standard would prevail as it has been nationally adopted and is necessary for achieving compliance with abattoir licence conditions. Nevertheless, given this apparent inconsistency in both sentiment and state and the territory application of these two documents, public confusion on the regulation of religious slaughter is perhaps unsurprising, particularly when these practices occur in a global context where Australian norms differ from those in many other countries.

### 3.2. Export Abattoirs

In contrast to domestic abattoirs which are regulated under state law, abattoirs supplying meat for export fall under Commonwealth (Cth) jurisdiction by virtue of the trade and commerce power of the Constitution [40]. Much of Australia’s export income from livestock trade comes from the live export of animals with slaughter occurring in the receiving country [91]. As a result of several key events accompanied by considerable media coverage, this trade has evoked ongoing controversy and predominantly negative attitudes towards the trade amongst the Australian public [92]. Key events include the 2003 Cormo Express incident, mistreatment of Australian cattle in a Cairo abattoir in 2006, the 2011 Four Corners broadcast, showing cruelty to cattle in Indonesian abattoirs, and the 2018 Awassi Express disaster, where 2400 sheep died due to extreme heat as a result of being transported during the northern hemisphere summer [93]. It is likely that media images of overseas abattoir cruelty and abuse, which tend to depict traditional halal slaughter, have promoted negative attitudes towards halal methods in Australia, and may contribute to some misunderstandings around domestic practices. The complex regulatory framework surrounding live export is outside the scope of this review. However, as there is an overseas market for meat from animals slaughtered in Australia, the following describes the regulation of this market. 

Abattoirs that export meat products may operate under a Tier 1 or Tier 2 arrangement. Tier 1 arrangements allow for the supply of products to markets that accept product produced in accordance with the “Meat Standard”. Regulatory oversight is provided by the state regulatory authority, as listed in Table 1. Tier 1 markets include inter alia Egypt, Kuwait, and Indonesia. Halal products must also be certified under the Australian Government Authorised Halal Program (AGAHP). Tier 1 operations are registered under the Export Control Act 1982 (Cth) [94] and operate under an Approved Arrangement issued under Schedule 2 of the Export Control (Meat and Meat Products) Orders 2005 (Cth) [95] or the Export Control (Wild Game Meat and Wild Game Meat Products) Orders 2010 (Cth) [96]. Essentially, this regulatory framework requires compliance with the Meat Standard and therefore all slaughter must involve pre-stunning. After an initial verification process, the Australian Government Department of Agriculture and Food delegates enforcement activities to the state regulatory authority. The latter is charged with ensuring compliance with the Meat Standard, with the Federal department providing oversight through assessment of state or territory control systems [97]. 

Tier 2 export markets require registration and oversight of facilities by the Federal government. Tier 2 accreditation allows access to all export markets, including the EU. This is primarily achieved by employing Australian Government Authorised Officers (AAOs) and in-plant dedicated veterinarians (Veterinarians-in-Charge) to carry out meat inspection. In addition to the Australian Meat Standard, facilities must meet the requirements of the Export Control Act 1982 and any additional requirements of the importing country [98]. A key feature of the regulatory regime for both Tier 1 and 2 abattoirs is the need for adherence to the Meat Standard. As discussed earlier, this requires pre-stun for all slaughter except in exceptional circumstances. Furthermore, the Australian Quarantine and Inspection Service (AQIS) has developed specific guidelines [99] issued pursuant to the Export Control Act 1982 which regulates the slaughter of animals for halal export purposes. Standard 6 requires that animals must be effectively stunned before sticking. Based on this network of delegated legislation, the public should be assured that pre-stunning of animals is very much the norm in Australia for halal export products. 

### 3.3. Non-Commercial Slaughter

Whilst the slaughtering of animals for personal consumption is likely to be extremely limited in terms of animal numbers and persons engaged in the practice, a brief foray into the relevant law is warranted, particularly as it is frequently raised by opponents of halal practices. The state Meat Act regulatory structure considers such a scenario and provides an exemption from requirements for accreditation, and therefore from need to comply with the Meat Standard. For example in South Australia, the Primary Produce (Food Safety Schemes) (Meat) Regulations 2017 [57] provides that accreditation does not apply to:(a) the growing of poultry, the killing of an animal, or the further processing or handling of an animal, at premises by or on behalf of the owner of the premises if none of the meat from the animal is—
(i)sold or intended for sale; or(ii)used, or intended for use, as food for paying guests; or(iii)taken away, or intended to be taken away, from the premises.


Under the Animal Welfare Act 1985 (SA) [80], ill-treatment for the purposes of defining the cruelty offence is said to be:s13 (3)(g) kills the animal in a manner that causes the animal unnecessary pain; or(h) unless the animal is unconscious, kills the animal by a method that does not cause death to occur as rapidly as possible.

Taking these two provisions together, if a person were to perform religious slaughter (without stunning) on an animal for consumption by themselves and their family, and they were able to do this without causing the animal unnecessary pain, they would not be in breach of the law. Should a matter come before the court, the outcome would likely depend on judicial discretion based on interpretation of "unnecessary" with reference to case specifics and witness testimony. To the authors’ knowledge, there is no published Australian case law on this topic. Such a case would likely be complex since it would expose the tension between freedom of religion which is expressed in international law [100], and established science on the risks to welfare from non-stun slaughter. 

## 4. Perception of the Australian Public

The requirements of kosher and halal slaughter from both religious law, and regulatory perspectives, are poorly understood by most Australians. For example, most halal-certified abattoirs in Australia use reversible stunning, but a 2016 study found that the majority of Australians surveyed believed that stunning is never allowed in halal slaughter [11]. A similar lack of understanding of the facts and requirements for halal slaughter was also found in the UK amongst veterinary students, a group that we might expect to be well-informed [101]. Australians were also likely to believe that Imams must approve all halal meat, and that the slaughterer must completely sever the animal’s head, neither of which are required, with the latter being expressly forbidden [11]. The same study also found that the small number of Jewish participants in the study were opposed to the slaughter of conscious animals for religious purposes, despite the fact that stunning is not permitted under shechita. This seeming anomaly may indicate a lack of knowledge about kosher slaughter practices among Jews, but it may also be a factor of the relatively low adherence of Australian Jews to strict kosher practices. This latter point is significant because of the disparity in public scrutiny regarding halal and kosher slaughter. A search of Australian newspapers over the past 5 years (28 February 2015–28 February 2020) in the database ProQuest International News stream returned 727 results for the term “halal” but only 214 results for the term “kosher”. This difference may be partially attributed to the numbers engaging in each type of practice or perhaps the association between halal slaughter and live export [102], but it is likely that Islamophobic attitudes also play a role given their increased incidence in Australia generally [8]. As Thomas and Selimovic observed in their Norwegian-based study, much “coverage of halal in…newspapers is mediated through an Islamophobic lens” which contains “deliberate linkage of halal with crime, terrorism and anti-Muslim political figures” [103]. Further analysis of media coverage of religious slaughter in Australia is warranted to better understand how it shapes public discourse in Australia.

In a 2016 survey, most Australians expressed the belief that quality of halal meat was the same or lower than non-halal meat [11]. Most Australians also believed that slaughtering conscious animals for religious reasons was unacceptable and reported that they chose to avoid purchasing halal meat, most often for animal welfare reasons. A small but vocal group of Australians have taken more extreme measures, including organising boycotts and submitting anti-halal perspectives to a parliamentary inquiry focused on certification more generally [104]. Some anti-Islam groups have claimed that the increase in halal certification by mainstream Australian food producers and retailers is a financial burden on Australian consumers, and even that halal certification fees are a "religious tax" that support jihad efforts, but there is no evidence behind either of these claims [105]. Bullying on anti-halal social media pages has had deleterious impacts on some businesses, such as the South Australian dairy company Fleurieu Milk which succumbed to pressure to relinquish its halal certification after such pressure and lost a lucrative contract with Emirates Airlines [104].

One telling indication of anti-halal sentiment in Australia is the hijacking of a federal parliamentary inquiry into a more general exploration of third-party certification of foods. Upon accepting public submissions, the inquiry was inundated with submitters expressing concerns and/or opposition regarding halal certification. While some expressed what the committee considered to be valid arguments, a number were described by the committee as “inflammatory, derogatory, and in some cases, even obscene”. The main objection of most submitters was the insufficient labelling of halal-certified food, which they wanted to avoid purchasing because they did not wish to contribute to “promoting and funding religious practices”. Another submitter described halal certification as a “religious tax that funds the growth and spread of Islam” and “a way of imposing sharia law and Islamic religious beliefs”. Yet another expressed concern that halal certifiers were “funding other organisations that have strong links to criminal activity such as Terrorism”. Finally, some submitters expressed animal welfare concerns regarding the perceived lack of stunning and partial stunning, with some referring to religious slaughter generally and others specifically to halal slaughter [105].

Because both Islamic and Jewish law mandate specific slaughter practices that are accommodated in Australia under the auspices of freedom of religion, it is important to scrutinise the public focus on the former but not the latter. Fischer [106] argues that, in a global context, halal markets and certifications are currently subject to widespread attention because they are in the midst of a process of establishing authority and credibility, whereas kosher markets are well-established and more settled. This perspective recognises the state of flux of halal certification and regulatory practices in a landscape characterised by growing Muslim diaspora communities, increased production of halal food “at a distance” through complex systems of global food provision, and debate among Islamic scholars and followers about what practices, slaughter and otherwise, are halal and haram. However, in the Australian context, we must also recognise significant differences between the Muslim and Jewish populations in terms of their histories, patterns of population growth, longevity and continuity of communities, religiosity, and socioeconomic status. Jews may simply be less visible than Muslims. The focus on halal (and relative lack of concern about kosher slaughter) is thus arguably an example of “culinary xenophobia” [107] by which “…halal becomes an indicator or litmus test of the West’s un/willingness to accommodate and integrate Muslims” [103].

## 5. The Appropriateness of the Regulatory Regime

### 5.1. Fragmentation in a Decentralised System

It is clear from the above that the regulatory regime for religious slaughter is divided between the Commonwealth and the states depending on the final destination of the meat product. This duality leads to a regime comprising an array of statutes, regulations, orders and standards. The overall objective of the law should be to provide a framework that is easily able to be monitored and enforced. However, the presence and intersection of different enforcement agencies at both state and federal level creates challenges to enforcement, as well as public transparency. Whilst setting uniform legislation is challenging in our federal system of governance, there are mechanisms that can be considered for achieving greater consistency across the states and territories. A strategy frequently cited is via Commonwealth leadership in setting standards to enable a consistent approach to legislation. This was the original intention of the Australian Animal Welfare Strategy, whose purpose was [108]:
…that streamlined, efficient, transparent and successful processes are developed to deliver nationally consistent animal welfare outcomes…. the benefit is that there is improved effectiveness and efficiency in processes and the application of resources to develop and implement animal welfare policies and systems.

The proposed strategy also allowed for wider representation and oversight provided through the Advisory Committee to AAWS [109]. This move seemed to reflect the political nature of decision-making in the animal welfare space which counteracted some concerns that policy was largely driven by industry [110]. The AAWS advisory committee was disbanded due to budget constraints in 2013 [111]. Since this event there appears to have been little progress in furthering the AAWS mission statements. 

In 2017, a productivity commission report [112] focused on the regulation of the agricultural industries suggested several failings of the current system:Current standard setting failed to consider community values and expectations about animal welfare;There was a lack of incorporation of science into the standard setting process;That conflicts of interests existed between state and federal agricultural departments when managing animal welfare.

A key outcome of the report was a suggestion for a statutory agency to develop national animal welfare standards with the incorporation of science and community engagement. This suggestion was not new, with a similar idea having received cross-partisan political support previously. The Greens introduced a bill to establish an independent statutory authority in 2015, and the ALP made a pre-election promise in 2016 to create such an office. The coalition elected to not support the national proposed body, citing concerns about cost to the commonwealth by requiring enabling legislation, relocation costs and hiring of specialist expertise, with an accompanying additional layer of bureaucracy [113]. The recommendation has been reignited on occasion following an animal welfare scandal but does not appear to have received political goodwill with the current administration. Structural change seems unlikely; however, at the very least it would be valuable to resolve the inconsistencies of language between the various documents as highlighted above. 

### 5.2. Use of Delegated Legislation

The legal framework in relation to slaughter places a heavy reliance on delegated legislation and other instruments, such as codes. This raises two main concerns. First that they are of uncertain legal status and subject to significant discretion in interpretation and application. Secondly, the document authors are frequently government agencies and industries who arguably possess a conflict of interest due to having interests which are not primarily about animal welfare but tend to be economically motivated [114]. Furthermore, these documents have not gone through the parliamentary law-making process, with oversight assumed to have been provided through the process of “tabling” the legislation for review and an opportunity for disallowance. The sheer volume of documents of this form, however, decreases the efficacy of this process, with parliament perhaps not even notified of the existence of the document for review [114]. The further safeguard of the requirement to have a regulatory impact statement which considers public input is also removed for minor changes to delegated legislation and some codes [114,115]. It is not impossible to envisage a model where key provisions relating to religious slaughter currently contained in the standards are included directly in the state meat/food regulations (as occurs under the UK regime [116]). This reform would assure increased parliamentary oversight and encourage transparency to the public. In fact, it is surprising, given the public interest, as well as the constitutional protection for freedom of religion embodied in the issue, that such a reform has not been proposed. 

Associated with the use of delegated legislation, including industry-created standards, is the creation of a regulatory model which is essentially that of self-regulation in standard setting, compliance checking, and enforcement [90]. This model is frequently criticised in jurisprudence as it allows subversion of rules for private purposes due to possession of information control. It has been suggested that such systems poorly protect the public interest and confer a lack of transparency [117]. However, this model has been reported to be the result of an increasingly common trend in “regulatory evolution of animal welfare governance” in Europe [118]. This trend has seen commercial entities, NGOs, and industry bodies taking on matters of societal concern and are not limited to animal welfare but occur in food safety and quality, forestry and fisheries governance. A number of explanations for this development have been proposed, including growing disquiet with government policy making [119], increasing consumer demand and the need for adequate response, and isomorphism where similarities in structure and processes result either by design or as a result of the external political environment [120]. Advocates of this model suggest these new governance tools of voluntary agreements, partnerships and delegated law are more effective, participatory and less hierarchical than traditional legislative means [121]. In legislative terms, there is a shift both in form and authority of the relevant legal framework. Firstly, as opposed to the traditional position where a breach of public animal welfare standards becomes a matter for resolution by criminal or administrative law [122], it becomes a breach of a voluntary contractual agreement. Secondly, instead of the use of state authority to encourage compliance, market authority is the incentive. Proponents of the private governance approach claim that voluntary commitment increases compliance and, therefore, welfare outcomes, and brings with it the ability to rapidly adapt to changing conditions [123]. These governance mechanisms can also transgress state boundaries, encouraging globalisation and enabling a means of regulating overseas production practices where governments can have no influence [124]. This has particular relevance when considering the scenario with regard to meat or live animal export.

### 5.3. Transparency of Practice

Transparency in relation to animal welfare is related to the visibility of production practices and their regulation to all stakeholders, including the public and government. With increasing awareness of animal welfare, product consumption can be influenced by public understandings, as can attitudes toward relevant stakeholders. It is also essential that parties understand how practices impact on animal welfare [125]. Typically, commercial, cultural, and attitudinal factors have limited bi-directional communication between the agricultural industry and consumers resulting in some confusion as to the nature of modern farming practices [125]. Issues of animal welfare, food quality, and sustainability have received increased consumer attention, and with increased use of the internet and social media—including information from other countries—information (or misinformation) is readily available to consumers. The agricultural policy agenda therefore needs to consider consumer and broader community engagement as a priority. The public may be hesitant to engage with the details of processes associated with slaughter, given what has been termed the "meat paradox", which is a state of cognitive dissonance arising from the fact that many people like to eat meat but do not want to think about or be connected with the morally troublesome aspects of it, particularly the killing of animals [126]. Thus, it is critical to focus on robust regulation and oversight, as many consumers and community members will rely on these processes to ensure that animal welfare is safeguarded.

The fragmented nature of the legal regime, as well as minimal provision of public data about administrative or enforcement activities from government agencies [127] and the easy availability of information from a variety of locales outside of Australia with different legal and practice regimes, has created an information gap (accompanied by proliferation of misinformation). Thus, the current system tends to mask the extent to which animals are protected by law and potentially leads to community mistrust of enforcement agencies and adequacy of current regulations, which is antithetical to the principles of government openness and accountability [127]. 

## 6. Conclusions

Further qualitative research on public attitudes to, and misapprehensions of, religious slaughter practices is needed. As documented in this paper, greater transparency and improved understanding of current slaughter practices and regulation is likely to result by having more clear and consistent legislative provisions as well as increased independence from industry in the setting of standards, enforcement, and administration. A starting point for legal reform would be the relocation of important provisions pertaining to religious slaughter from delegated codes to the responsible act or regulation, ensuring proper parliamentary oversight. In addition, more active public engagement must occur, particularly with regard to what constitutes legal practices and animal welfare standards in the Australian context to overcome ongoing conflict between those who oppose religious slaughter and the Muslim and Jewish communities.

**Table 1 animals-10-01530-t001:** Legislation in the Australian states and territories that allows religious slaughter to take place, and mechanism of legal protection/safeguards for the practice.

State/Territory	Animal Welfare Legislation			Meat Industry/Food Acts		
	Legislation	Sections/Nature of Implementation	Enforcement Agency *	Legislation	Sections/Nature of Implementation	State Regulatory Agency
Australian Capital Territory	Animal Welfare Act 1992 [76]	General protection via s 6A—definition of cruelty contains the qualifying words unjustifiable, unnecessary or unreasonable.	RSPCA ACT [128]	Food Act 2001 [129]	Food Regulation 2002 [53]. There is no reference to the Standards in the legislation. There is no abattoir in ACT [130].	ACT Health Protection Service
New South Wales	Prevention of Cruelty to Animals Act 1979 [77]	s 24(c) provides specific referral.	RSPCA NSW [77] Animal Welfare League NSW [77]	Food Act 2003 [46]	The standards are incorporated under reg 83 of the associated Food Regulation 2015 [54].	NSW Food Authority
Northern Territory	Animal Welfare Act 1999 [78]	s 79 provides a defence to prosecution if an element of the offence, was in accordance with cultural, religious or traditional practices.	Department of Primary Industries and Resources [78]	Meat Industries Act 1996 [52]	Reference to standards on issued abattoir licence. Regulation 10 of the Meat Industry Regulations 1997 [55] requires compliance with Standards listed.	Department of Primary Industry and Resources
Queensland	Animal Care and Protection Act 2001 [79]	Religious slaughter is exempted as an offence in s 45.	RSPCA Queensland [131] Biosecurity Queensland [132]	Food Production (Safety) Act 2000 [49]	Listed in Sch 10 of the Food Production (Safety) Regulation 2014 [56] as an "advisory standard" which in spite of as otherwise suggested by wording must be complied with (see Regulation 5).	Safe Food
South Australia	Animal Welfare Act 1985 [80]	General protection terms such as "unreasonable" and "unnecessary" in s 13. Model Code of Practice [88] prescribed via Schedule 2 of the Regulations [62].	RSPCA SA [133]	Primary Produce (Food Safety Schemes) Act 2004 [47]	Schedule 1 of Primary Produce (Food Safety Schemes) (Meat) Regulations 2017 [57] lists standards. Penalty in event of breach set out at Regulation 11.	Primary Industries and Resources SA—PIRSA (Biosecurity)
Tasmania	Animal Welfare Act 1993 [81]	General protection by use of terms such as "unreasonable" and "unjustifiable" at s 8.	RSPCA Tasmania [134]	Primary Produce Safety Act 2011 [51]	Standards listed in Sch 2 of the Primary Produce Safety (Meat and Poultry) Regulations 2014 [58]. Regulation 12 provides that Schedule 2 listed Standards to be complied with.	Biosecurity Tasmania
Victoria	Prevention of Cruelty to Animals Act 1986 [82]	S9(1) General protection at s 9(1) through use of term "unreasonable".	RSPCA Victoria [82] Department of Job, Precincts and Regions [82] Department of Environment, Land, Water and Planning [135]	Meat Industry Act 1993 [48]	Standards incorporated by direct reference throughout Meat Industry Regulations 2015 [59].	PrimeSafe
Western Australia	Animal Welfare Act 2002 [83]	General defence to cruelty provided by s 22 if authorised by or under a written law (such as Meat Industry Authority Act) and done humanely.	RSPCA WA [136] Livestock Compliance Unit [83]	Food Act 2008 [50]	Regulation 18 (1) of Food Regulations 2009 [60] adopts Standards. Regulation 46 makes compliance with standards compulsory and non-compliance subject to penalty.	WA Health Department (abattoir licences granted by Australian Meat Industry Authority under Western Australian Meat Industry Authority Act 1976 [137])

* All state and territory police forces are given powers to enforce the animal welfare legislation. The act, which allows for the adoption of the Standards (Australian Standard for the Hygienic Production and Transportation of Meat and Meat Products for Human Consumption and related documents pertaining to other species) has been cited. There may be other acts with relevance to processes occurring in abattoirs.

## Figures and Tables

**Figure 1 animals-10-01530-f001:**
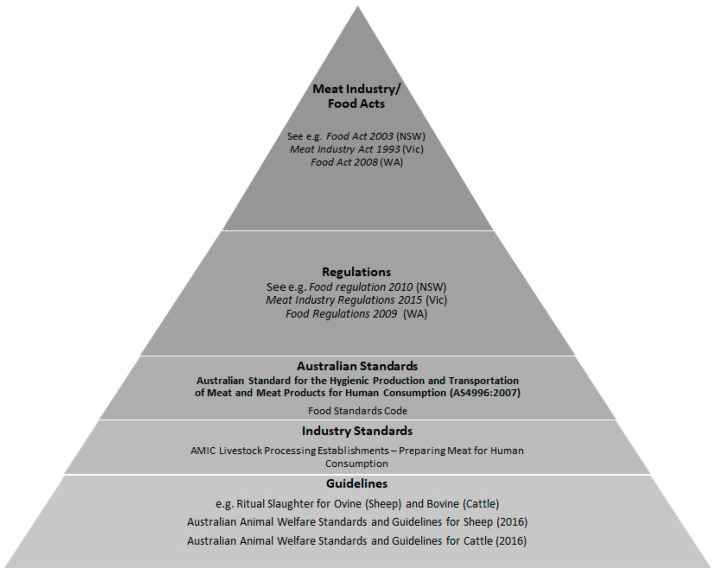
Hierarchy of state and territory legal regime in relation to religious slaughter. Note that standards and guidelines listed are illustrative and do not represent a comprehensive list.

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
