# Peer review of "A Review of Legal Regulation of Religious Slaughter in Australia: Failure to Regulate or a Regulatory Fail?"

_animals, 2020, doi:10.3390/ani10091530_

Round 1

Reviewer 1 Report

Legal Regulation of Religious Slaughter and the Australian Community: Failure to Regulate or a Regulatory Fail?

Loyer, J. et al.

My compliments to the authors.  I appreciate the review of differences between Halal and Shechita slaughter methods and the confusion that has developed regarding regulatory oversight.  I find very little in terms of concerns with exception of a couple of minor items. 

I notice you refer to Judaic method of slaughtering as “shecita” throughout the manuscript.  I believe the proper spelling is “shechita”.  Please check the spelling. 

Lines 343-345     This surely providing some legal recognition of the inevitable animal welfare impact of any method of humane killing or slaughter.

I suggest checking this sentence – I feel like it reads awkwardly.

 Line 529             The proposed strategy also allowed for wider representation and oversight provided though the Advisory Committee to AAWS.

Please check this sentence – I believe “though” should be changed to “through”

Author Response

I notice you refer to Judaic method of slaughtering as “shecita” throughout the manuscript.  I believe the proper spelling is “shechita”.  Please check the spelling.  

Thank you for bringing this to our attention. We have now ensured all instances are spelt correctly.

Lines 343-345     This surely providing some legal recognition of the inevitable animal welfare impact of any method of humane killing or slaughter.

I suggest checking this sentence – I feel like it reads awkwardly.

Agree- we have amended sentence to read “This clause highlights legal recognition of the inevitable animal welfare impact of any method of humane killing or slaughter”.

Line 529             The proposed strategy also allowed for wider representation and oversight provided though the Advisory Committee to AAWS. Please check this sentence – I believe “though” should be changed to “through”

Thank you- we have amended to ‘through’.

Reviewer 2 Report

See pdf attached

Author Response

This submission is a review of religious slaughter policy and regulatory issues in Australia. The title includes reference to the “Australian Community.” This might be regarded as a little ambiguous. Does it really refer to Australian society across the continent, the two religious communities, the animal welfarist community or the stock farmers who depend on an efficient and clearly regulated animal processing industry?

To avoid any possible ambiguity we have removed reference to the Australian community in the title. The title is now ‘A Review of Legal Regulation of Religious Slaughter in Australia: Failure to Regulate or a Regulatory Fail?.

Spelling is British or unusual and mixed with typos. There is potential for confusion here. Shechita NOT Shecita (three times), most prefer carotid to carotic, and oesophagus is archaic in my North American experience. The editor will have to advise these authors about British variant orthography.

I think we would prefer to stick with the British spelling given the authors’ backgrounds but will take this on editorial advisement. Thank you for pointing out the misspelling of Shechita in places- this has now been corrected.

Details of Jewish religious laws are a bit confusing: kashrut vs. halacha.

We agree that this section may confuse readers. Since we are discussing kosher meat and referring to Jewish dietary law we have now used ‘kashrut’ throughout.

 What does (Cth) mean? (see l. 397)

Cth is the standard abbreviation for commonwealth used for referring to legislation to distinguish it from the State and Territory Acts. We have included reference to this abbreviation at line 377.

The discussion of Islamophobia in Australia (ll. 71-80) strikes me as a little facile. I think that the authors should have a closer look at this literature.

We accept that there is potentially more room for discussion on this concept but this is not the purpose of the article. We also feel that to do justice to this discussion there would need to be more empirical data in the Australian context expressly linking (or not) halal products and islamophobic sentiment.

Carbon Dioxide stunning (ll. 153-158) is only used with pigs or poultry (not noted) and the information seems to rely entirely on one passage from Animal (9:2). “Indicators used in livestock to assess unconsciousness after stunning: a review” by M. T. W. Verhoeven et al. (2015). The authors paraphrase the passage quite closely, describing “immersion in a mixture of gasses consisting of (low level) oxygen (O2), carbon dioxide (CO2), argon (Ar) and/or nitrogen (N2).” Of course, the mixture they are describing sounds a lot like air while the whole point of this method is asphyxia in a mixture that features CO2 with some air that leaks in. I would like the authors to consult some technical literature on the gaseous composition of the CO2 stunning mixture.

We appreciate the reviewer’s comment and agree that without detailing percentage composition this could represent normal air. Given the focus of this paper is not on the stunning methods, and the purpose of this section is to provide an overview to set up the legal analysis, we have removed this level of detail. We now just refer to the “gasses creating an anoxic environment’.

CO2 stunning is described as non-Islamic (ll. 180-183) but this misses the whole point. CO2 stunning is mainly used for pigs and of course swine are forbidden by halal adherents.

The reviewer is of course correct and we have not made this point clear. We have now included that gaseous stunning is mainly used for pigs and poultry and tried to bring the point across that this section is referring to slaughter of poultry by gaseous methods.

On ll. 200-202, we are told that the authors have little empirical data on public perceptions and understanding of slaughter, religious practices and the law in Australia. Given the goals of the paper, I don’t believe that this is a problem that the authors can simply apologize their way out of. For this paper to make an academic journal contribution, it needs some reliable primary data.

We agree that there is a real need for some primary data. However, do not believe we can add this at this time without an avenue for funding so we can acquire robust, and representative data from the Australian public via survey or focus group methods. This paper was intended as a review to identify gaps for future research and we have clearly found some gaps.

 Ll. 211-323 summarizes the regulatory structure germane to the slaughter of food animals in domestic abattoirs. A Canadian or American would be interested to learn that state level inspection appears to apply to all domestic shipments, even those that cross state boundaries though the issue of interstate meat shipments is not specifically addressed. Apparently the legislation in each state then cites an Australia-wide ‘Meat Standard’) bringing the Commonwealth government into the legal regime. There are some interesting contradictions between different codes, standards and regulations and apparently both within and between states. These are interesting observations. But the next logical step would be to do some key informant interviews to learn more about what factors informed and drove the legislative and regulatory process and why these contradictions exist.

Agree- this would be a fantastic idea for future research but not one we feel we can pursue at this time without further resourcing.

 Slaughter is conducted for domestic, export and non-commercial purposes. Why not make these three rows in a table with columns for each food animal species and give estimated numbers (with percentages) for each?

These figures are not available publically in a form broken down by purpose and % halal/shechita compliant. The only figures we are able to source are animal numbers slaughtered for meat separated by species.  Deriving specific figures on halal/shechita slaughter would likely need primary study or Freedom of Information request.

“Delegated legislation” and occasionally “delegated codes” is clearly an important political concept which appears throughout the paper starting with the abstract but it needs to be explained in greater detail, especially considering the audience of Animals.

Yes- this terminology is perhaps not understood by those with mimimal legal training.  We have added a few sentences from line 218 to explain the ‘delegated’ terminology. The following is now included “These Acts empower responsible officials to make subordinate (delegated) legislation covering issues that fall within the scope of the Act. Delegated legislation includes regulations, ordinances and Codes of Practice/Standards. These documents contain more detail that the primary Act but are usually associated with reduced penalties in the event of breach of a provision, and may not be subject to parliamentary oversight.”

 I want to commend the authors for making progress on an important issue of food production and food policy and as an example more broadly of the challenge of the role of the state in relation to the public interest in regulating free enterprise. However, I am recommending against publication at this time. Why is it not ready for publication? It does not yet make the kind of contribution that is required for an academic journal because of its reliance on secondary sources for data, the lack of any unique application of theory, and the absence of a clear policy recommendation/implication for progress in this area.

 In addition, Animals is an international journal and I am sorry to say that the approach taken here seems quite Australian centered. What could the researchers do given the investment that they have made so far? 1. Run a survey at the state or national level and measure the attitudes and misapprehensions that are described too briefly at ll. 200-202 and again at ll. 450-511. Your title alludes to the “Australian Community,” more data is needed to characterize this community, however it is defined. 2. Interview some key informants at both the state and commonwealth levels to gain a better understanding of the reasons for the various regulatory curiosities that are catalogued in Section 3 of the paper. 3. Interview some key informants at the abattoir level (plant managers and meat inspectors) to convey a richer and more vivid impression of what works and what does not in the slaughter and processing of the various livestock species. 4. Assemble some descriptive statistics to give a sense of the numbers of species, numbers killed and exported by type of slaughter and for the three markets (domestic, export, and non-comercial). 5. Apply some innovative theoretical perspective to provide an alternative interpretation of these facts. Risk society? Bureaucracy theory? Neoliberalism? 6. Look at slaughter practices the regulatory environment in some other major livestock producing regions to understand what makes the Australian situation unique and where Australia might profit from emulating best practices overseas. The obvious question that I must ask is why is there no “Australian Food Inspection Agency”? The authors might take a look at Farm to Fork: A Strategy for Meat Safety in Ontario (2004). Any two of these suggestions would help the authors get to the sort of contribution that is required. I hope that they will persevere.

We agree that all of these suggestions would make for valuable study topics and we will certainly consider them for future work if resources become available to perform the work to a rigorous standard. We feel that trying to perform this without some research support would lead to poorly conducted, and likely non-representative sampling with the potential to mislead. This paper is presented as a review article to summarise the relevant legal framework and to suggest avenues for future research. We believe we have achieved this goal.

We have included some reference to theory in framing the governance shift from state-centred arrangements to increased use of self-governance and private actors. We believe this inclusion has created a more balanced argument describing potential reasons for this shift and the reported benefits of it. We feel that inclusion of extensive philosophical and political science rhetoric is outside the original scope of the article and cannot be backed by empirical data from the Australian jurisdiction.  We believe that our review paper contributes to the broader discussion as the first overview of the Australian regulatory framework pertaining to religious slaughter, highlighting the challenges brought about by a federated system.

Reviewer 3 Report

a clear and well written observation about th current stauts of religious slaughter in Australia and attitudes around such.

extensive refs inc legislation and basis of law making are included. while other countries can draw some comparisons, it relates directly and only to Australia due to unique legal framework of state/federal gov and the underpinning powers/responsiblites.

Some recent literature from Fuseni et al in the UK/EU has some similar observations, as has the EU DIAREL project- these could have been made more ref to as to broaden the wider appeal, especailly since the EU has agrugably the greatest challenges across its MS and has invested heavily in exploring attitudes and legal frameworks.

Author Response

A clear and well written observation about the current status of religious slaughter in Australia and attitudes around such.  

extensive refs inc legislation and basis of law making are included. while other countries can draw some comparisons, it relates directly and only to Australia due to unique legal framework of state/federal gov and the underpinning powers/responsiblites.  

Some recent literature from Fuseni et al in the UK/EU has some similar observations, as has the EU DIAREL project- these could have been made more ref to as to broaden the wider appeal, especailly since the EU has agrugably the greatest challenges across its MS and has invested heavily in exploring attitudes and legal frameworks. 

Thank you for your comments. We have included reference to the Fuseini et al 2019 paper which backs up our suggestion that there is community misunderstanding of religious slaughter law.

Round 2

Reviewer 2 Report

See PDF file

Author Response

In the second sentence of the simple summary (ll.13-14), the authors state: “The requirements of domestic religious slaughter practice, including animal welfare provisions, appear to be poorly understood by the Australian public.” (Emphasis added.) In the abstract (ll. 24-25), this becomes: “the details of religious slaughter practices, including related animal welfare provisions, are poorly understood by the Australian public.” The word “appear” is gone.  

This sentence has been amended to ‘However, the details of religious slaughter practices, including related animal welfare provisions, appear to be poorly understood by the Australian public…’ 

One of the three sources [45] is in Arabic and while the English text is quite interesting, it says nothing directly to substantiate limited public knowledge of halal. The government. The second source [44] is fascinating as an example of Aussie-style racism and Islamophobia which may lie at the root of the public misunderstanding of halal. I’d like to see this source [44] added to your detailed discussion of the inquiry on ll. 488-500. 

We are unsure which reference the reviewer is referring to here, ref 45 is an online article (in English) of an interview with Dr Mohammad Anas. The sentence referred to references this article and the Government report on halal certification. This is likely a confusion with tracked changes but we are unable to pinpoint the source referred to so have not added this to the section discussing the inquiry.  

Finally, you refer to a 2016 survey in the sentence ending at l. 459. This should also reference [11]. 

Our apologies, the reference had become lost on previous amendments. This refers directly to ref 11.  

Kasrut is misspelt!  

Thank you. The misspelling has been corrected at line 52.  

Figure 1 says nothing explicitly about poultry. I would recommend that the reference in the text and the figure caption be made more specific about the limitation of this figure (sheep and cattle) or that legislation, standards and guidelines for poultry and swine be added.  

This figure is designed to illustrate general hierarchy of the legislative structure. It does not provide a comprehensive listing of the guidelines for any species. We have added a note to make this clearer which now reads ‘Note standards and guidelines listed are illustrative and do not represent a comprehensive list. 

 “For example use of the words ‘must’ and ‘should’ abound, which to the ordinary layperson are often regarded as synonymous.” As a North American, I strongly disagree. The distinction between must and should is very common knowledge! I suggest that you find a better example to substantiate your claim of ambiguity. 

Upon further reading it appears that there may be some differences in interpretation of these words across jurisdictions with drafters in Canada, Australia and Britain tending to avoid the words 'should' and ‘shall’ due to a differing everyday usage of the word. The statement as written was however incorrect. The main issue as you point out is in ambiguity around obligation. Common law countries, espoused in drafting guidelines, tend to avoid use of the term for this reason and stick with ‘must’. A note to this effect has been added at lines 362-364.  

The discussion on ll. 502-516 might be well informed by a reading of Dorothee Brantz’s work: Brantz, D., 2002. Stunning bodies: animal slaughter, Judaism, and the meaning of humanity in imperial Germany. Central European History, 35(2), pp.167-193.  

This is a very interesting article but does have a limited focus being German, Jewish and with a historical focus. We thank the reviewer for bringing this to our attention but do not feel the need to directly reference this article in this section. 

Please add a sentence or two to explain the “meat paradox” on l.617.  

A sentence on the meat paradox has been added at lines 623-35. This now reads “The public may be hesitant to engage with the details of processes associated with slaughter, given what has been termed the ‘meat paradox’, which is a state of cognitive dissonance arising from the fact that many people like to eat meat but do not want to think about or be connected with the morally troublesome aspects of it, particularly the killing of animals [126].” 

 Have a look at the principles of capitalization. Phrases like “state law,” “commonwealth jurisdiction” and the constitution have no need for upper case capitalization. Many more examples could be cited: “under a Tier 1 or tier 2 arrangement” (l. 395). But be sure to capitalize “norwegian” [sic] in your references.  

Thank you for pointing this out. There are a number of instances where we have not been consistent in capitalisation especially with regard to ‘state’. We believe we have corrected these instances.  It is our understanding that when referring to the Constitution of Australia this is capitalised as this refers to the name of an Act, “the Commonwealth of Australia Constitution Act”.  Commonwealth is also usually capitalised when it refers to the Commonwealth of Australia. On checking, Tier 1 is capitalised in Government documents perhaps because it refers to a specific arrangement see eghttps://www.agriculture.gov.au/export/controlled-goods/meat/elmer-3/meat-establishment-tier1 

Norwegian has been capitalised in the references.   

Finally, I think that the conclusion could add one more paragraph on directions for future research, perhaps some ideas along the lines that I recommended in my earlier review 

Ideas for future research as provided in the previous review have been added to the conclusion (lines 634-9).